# Long-Term Cyclic Loading Impact on the Creep Deformation Mechanism in Cohesive Materials

**DOI:** 10.3390/ma13173907

**Published:** 2020-09-03

**Authors:** Andrzej Głuchowski, Wojciech Sas

**Affiliations:** Water Centre, Warsaw University of Life Sciences-SGGW, 02787 Warsaw, Poland; wojciech_sas@sggw.edu.pl

**Keywords:** cyclic loading, creep, ratcheting, failure, cohesive, Cam-Clay, undrained, shakedown

## Abstract

Long-term cyclic loading is observed in a wide range of human activities, as well as in nature, such as in the case of ocean waves. Cyclic loading can lead to ratcheting which is defined as progressive accumulation of plastic deformation in a material. Long-term cyclic loading causes a time effect (creep), which is a secondary compression effect. In this article, we conducted 15 triaxial tests on four types of cohesive materials in undrained conditions to evaluate the damage and failure mechanism. To characterize the strain and pore pressure development, we modified the Yanbu resistance concept. On the basis of the static creep tests, we concluded that the stress paths for undrained creep behavior have to take into account the pore pressure developed during long-term cyclic loading. Pore pressure build-up and plastic strain accumulation during long-term cyclic loading are dependent on the number of loading cycles. Finally, we proposed the failure criterion, which was based on the Modified Cam-Clay constitutive model.

## 1. Introduction

Over the past years, the effect of cyclic loading on the performance of soil structures has been a focus of numerous papers, mainly because of the development of road networks and an increase of human activities in zones where soft soils are subgrade soils. Therefore, the focus for testing has shifted from a dynamic to a quasi-static perspective, where permanent deformation is the main adverse effect [1,2,3,4]. Previously conducted cyclic loading tests on these types of soils have adopted a two-way manner, and for non-cohesive soils, the liquefaction has been studied extensively [5,6,7,8].

The impact of cyclic loading on soil response has the most destructive effect in undrained conditions. Excess pore water pressure decreases the effective stress in the soil skeleton, and therefore failure can occur [3,9,10,11,12].

Failure also develops after numerous load repetitions, due to the accumulation of irrecoverable deformations; as a result of plastic deformation accumulation, the soil reaches the serviceability limit. Therefore, recently, deformation growth during cyclic loading has attracted more attention [1,4,13,14]. Accumulation of deformation phenomena have been divided into a few categories. The most important limit is between the deformation behaviors, i.e., which will lead to failure and which will be safe for the construction. The most recent deformation category is the shakedown concept [14,15,16,17], first applied to the categorization of unbound granular materials deformation [18,19,20,21]. The shakedown concept states that the plastic deformation during long-term cyclic loading cases can fail, creep, or shakedown, where the plastic strain rate decreases, and finally, the material achieves an elastic response. Numerous research papers, in cases concerning non-cohesive soils, have reported observation of the shakedown phenomenon.

Furthermore, shakedown criterion for sub-base soils have been adopted [22,23,24,25,26], and recent investigations have reported a transition zone between the plastic shakedown and plastic to creep, and have suggested narrowing down the plastic shakedown limit to avoid a metastable state [27].

The cohesive soils respond differently to cyclic loading as compare with non-cohesive soils, where plastic strain accumulation is observed even after long-term repeating loads. This phenomenon is called abation and is described as the state when plastic strains or deformation occur during cyclic loading with a decreasing rate, but the process continues as the cyclic loading progresses. Such soil reaction was reported first by Goldscheider and Gudehus [28] and was explored by Wichtmann [29,30]. The cohesive soil response as abation was reported by Głuchowski and Sas [31] and seemed to be a universal response of the cohesive material to cyclic loading. Abation has a wide range of plastic strain accumulation intensity. This observation raises a question about how long the abation takes and, most importantly, is ultimately ablation of plastic strain. Accumulated plastic strain during cyclic loading is close to the characteristics of creep. This idea is part of the shakedown theory where the stress state above shakedown limit is called the plastic cyclic creep zone in which the soil develops plastic strain and where the increment per cycle decreases almost to a constant level [32]. Nevertheless, an additional plastic strain can occur between a few cycles where no plastic strains are observed, primarily in undrained conditions [9,16,33]. This behavior shows that the cohesive material in undrained conditions remains in an instability state, and failure of the structure is dependent only on the stress state imposed on the soil structure.

Tests performed by Arulanandan et al. [34] on silty clay have shown that undrained creep occured in stress-controlled conditions, and pore pressure build-up was time dependent and structure dependent. Additionally, an undrained static creep produced a particular condition in which the soil failed because the soil remained in constant void ratio *e* and deviator stress *q* conditions. Therefore, based on the test results, the authors stated that the only way that the failure could be accomplished was for the stress state to move across the yield boundaries until it reached the critical state line at the appropriate yield surface for a given deviator stress *q.* A graphical representation of this conception is presented in Figure 1.

The undrained behavior of marine origin clay under cyclic loading was studied by Li et al. [11]. The test was conducted in a one-way manner, and the analysis of the results showed that a strong correlation existed between the secondary compression coefficient *C_α_* and stress state, which was described by the quotient of deviator stress *q_cyc_* and initial mean normal stress *p*_0_. The authors also pointed out that the undrained behavior during cyclic loading, which was referred to as a case of static undrained creep case, and all inelastic strains were viscoplastic, therefore, Perzyna’s overstress theory could be applied to calculate the viscoplastic strain rate.

Ni et al. [35] recently developed a cohesive material constitutive model for the case of cyclic loading. Their paper presented the concept of using Cam-Clay theory and a new elastic yield surface and modeled cohesive soil during unloading. The model needed two additional parameters to accurately simulate the cyclic loading effect on kaolin samples in a range of 1000 cycles.

In this article, we present the results of undrained one-way stress-controlled cyclic loading triaxial tests on three types of cohesive material. The tests were designed to understand clay behavior under long-term cyclic loading. Therefore, we conducted long-term cyclic loading tests. Moreover, we analyzed the data, and based on the static creep and resistance concept, we applied it to the cyclic strain development analysis.

## 2. Materials and Methods

### 2.1. Material Properties

We subtracted soil samples from three excavation sites of road construction in Warsaw. The soil was dried out, and then ground into powder. The first aim was to test the basic physical properties. The sieve and aerometric analysis led us to recognize this type of soil as sandy, silty clay (sasiCl), or clayey sand (clSa) CL and SC, according to the Unified Soil System (USCS-ASTM D2487-11). In Figure 2a, the soil gradation curve is presented. The result of the Proctor method test is shown in Figure 2b. The liquid limit obtained from the Casagrande apparatus W_L_, the plasticity limit W_P_, the plasticity index I_P_, and colloidal activity is presented in Table 1. The soils have a pale brown color, and this type of soil in the natural state has low to medium undrained strength ca. in the range of 20 to 60 kPa [36,37]. We tested the CaCO_3_ content with the use of a 20% HCl solution. The results of the test show that the soils have from 3% to 5% of CaCO_3_.

In this study, the soil samples were compacted with different moisture contents following the Proctor method. The compaction energy was equal to E_c_ = 0.59 J/cm^3^. The maximal dry density of the soils from the Proctor test were equal to 1.94, 1.92, and 1.98 g/cm^3^ and the optimum moisture contents, *m_opt_*, were equal to 12.18%, 12.54%, and 12.51%, respectively, (Figure 2b).

### 2.2. Sample Preparation

For cyclic triaxial tests, we prepared samples with a dedicated cylindrical metal mold with diameter, *d*, equal to 7 cm and with height, *h*, equal to 14 cm, which corresponded to triaxial sample dimensions. The pulverized soil was wetted to the required moisture, and then compacted using the Proctor method, where the energy of compaction was equal to 0.59 J/cm^3^. During the sample preparation, we measured the mass of the sample and the actual dimensions.

### 2.3. Loading Conditions

In this study, we adopted composite stress parameters as deviator stress *q* and mean stress *p*, which have been widely used in soil mechanics. In terms of principal stress parameters *σ*_1_, *σ*_2_, and *σ*_3_, the *q* and *p* are the stress invariants defined by Equation (1):(1)$$q=\frac{1}{2^{\frac{1}{2}}}{\left[{\left(\sigma_{1}-\sigma_{2}\right)}^{2}+{\left(\sigma_{2}-\sigma_{3}\right)}^{2}+{\left(\sigma_{3}-\sigma_{1}\right)}^{2}\right]}^{\frac{1}{2}}\;p=\frac{1}{3}\left(\sigma_{1}+\sigma_{2}+\sigma_{3}\right)$$

In typical compression tests, as this study, *σ*_2_ = *σ*_3_ and the equations are rewritten as Equation (2):(2)$$q=\sigma_{1}-\sigma_{3}\;p=\frac{1}{3}\left(\sigma_{1}+2\sigma_{3}\right)$$

In terms of the stress tensor, the mean stress is called the spherical part of the stress tensor, and deviator stress, *q*, is called the deviatoric part of the stress tensor. The stress parameters, *q* and *p*, are the total stress parameters which do not include the impact of pore water pressure, *u*, on the stress state in the sample. The saturated soil is a two-phase continuum material where the effective stress parameters are defined as in Equation (3):(3)$$q^{\prime }=q={\sigma^{\prime }}_{1}-{\sigma^{\prime }}_{3}=\sigma_{1}-\sigma_{3}\;p^{\prime }=\left(\frac{1}{3}\left(\sigma_{1}+2\sigma_{3}\right)\right)-u$$
where *p’* is often called the mean normal effective stress and *q’* = *q is* the deviator stress. During the compression tests, we observed a decreasing volume of the specimen. The change of the volume was possible mostly due to the porous nature of the soil material. In the case of the most unfavorable conditions, the voids were fully saturated (so-called undrained conditions) with water, and therefore the void volume remained unchanged during the compression test. In contrast, if the volume changed, and the pore water received a portion of compression energy called pore water pressure which in other saturation conditions was transmitted by the soil skeleton. The compression tests caused normal and shear stresses in soil specimens. The pore pressure did not affect the shear stresses. In Equation (3), the deviator stress value is the value of stress imposed on the sample, excluding the normal and effective normal stress state. The mean normal effective stress refers to the stress state in the soil, which makes one able to quantify the impact of deviator stress on the soil skeleton stress state.

The cyclic loading type was a one-way type where there was no reverse in the direction of loading. The frequency of loading *f* was equal to 1 Hz, and the parameters of cyclic loading can be described as presented in Figure 3. The one-way cyclic loading is conducted in constant mean deviator stress (*q_m_*) conditions with a deviator stress amplitude *q_a_*. Therefore, the maximal deviator stress applied to the sample is the sum of *q_m_* and *q_a_* and is denoted as maximum deviator stress *q_max_*. Analogously, the minimum deviator stress is the difference between *q_m_* and *q_a_*. The cyclic loading causes the axial strain presented in Figure 3b. The cyclic loading can also be presented as deviator stress-axial strain characteristic. The loading form is a sine waveform, and the mean deviator stress *q_m_* > 0 and the stress amplitude 0 < *q_a_* < *q_m_*. The mean deviator stress and deviator stress amplitude remained constant over the test duration. Therefore, the mean plastic strain and pore pressure development did not depend on the period of fast loading.

### 2.4. Test Apparatus and Test Procedure

In this study, we conducted cyclic triaxial tests using ELDYN GDS Instruments Ltd., the U.K. The pressure and volume controllers in this apparatus are automatic. The measurement of pore pressure took place at the top and the bottom of the samples with a pore pressure transducer, and in the case of soil Type 4 we used the mid-plane pore pressure transducer. The electro-mechanical loading system highly sensitive displacement transducer was responsible for the measurement of axial strain. The axial load was measured with an internal submersible load cell.

The test procedure consisted of a few stages, the first stage was a saturation step. The termination of saturation took place when the pore water pressure coefficient *B*-value was equal to 0.95 or higher. The second stage was an isotropic consolidation step. The end of isotropic consolidation conducted at the required effective confining pressure *σ’_c_* passed, when the pore water pressure effluent was less than 5 mm^3^ per five minutes.

After the consolidation stage, we carried out cyclic loading following procedures for undrained conditions. The cyclic loading was conducted in a one-way manner; in this study, we applied the sine wave with frequency *f* 1 Hz. The range of maximum cyclic stress *q_max_* was between 30 kPa and 40 kPa, and the effective consolidation pressure *σ’_c_* was between 18 kPa and 275 kPa. The physical properties of the tested samples and cyclic loading program parameters are presented in Table 2.

## 3. Results

The cohesive material reaction to cyclic loading was divided into pore pressure, axial strain, and effective stress state analysis categories to simplify the test result presentation. In Figure 4, an image of pre- and post-testing samples is presented.

### 3.1. Effect of Pore Pressure Development

In Figure 5a–d, the results of the excess pore water pressure Δ*u* (where Δ*u* = *u* − *u*_0_ and *u*_0_ is the pore pressure at the beginning of the cyclic loading triaxial test) under different cyclic stress conditions are summarized. The pore pressure increment is dependent on the initial consolidation pressure *σ’_c_*. The cohesive material, in this study, was compacted using the Proctor method. The oedometric tests showed that this type of compaction caused the preconsolidation state. The preconsolidation pressure is equal on average to 270 kPa [38]. In Figure 5a, we can see that the Δ*u* in the first cycle is equal to 20.0 and 16.5 kPa when the consolidation pressure *σ’_c_* is equal to 45 and 90 kPa. In contrast, the Δ*u* has a smaller value (12.5 to 12.9 kPa) for a greater *σ’_c_* (135 and 90 kPa, respectively).

During cyclic loading, the pore pressure rate is more significant for the samples consolidated with a pressure equal to 90 and 135 kPa. Notably, in the early stage, excess pore water pressure achieves most of its total value. The pore pressure after 1000 cycles has similar characteristics in the logarithmic scale. This indicates that the abation of pore pressure occurs. Therefore, we observe that after rapid pore pressure generation in the first few hundred cycles, it stabilizes. The stabilization of the pore pressure indicates that the samples response to cyclic loading can be described as hardening. We observed a similar response to cyclic loading in the case of sample A.01. In this case, the pore pressure rapidly increases to 18.9 kPa in the first cycle. Then, the excess pore pressure rate remains constant during the first ten cycles, and then its rate gradually decreases. The pore pressure characteristics for the second soil type are similar to the first soil type.

For the samples B.01 to B.03, which correspond with samples A.01 to A.03, the excess pore water pressure in the first cycle is five times higher and corresponds to a decrease in consolidation pressure. In the case of sample B.04, the soil was loaded with an amplitude that was two times higher (11.0 kPa), and as a result, the excess pore water pressure rose significantly in the first cycle. Additional loading cycles remained similar to those observed in samples A.01 and B.01, where abation was recognized. An interesting soil response to cyclic loading was found for sample B.05. After 3000 cycles, the peak pore pressure was achieved by the sample, and we observed the reverse of excess pore pressure characteristics (see Figure 5b).

To explain this phenomenon, we refer to Arulanandan et al.’s undrained creep proposal. The soil Δ*u* decreased the effective stress, therefore, the stress path touched the yield surface. Because of the undrained conditions, no volumetric changes could occur, and yielding was observed as a decrease in excess pore pressure. At this point, we assume that the yield surface is shrinking with the yielding of the material that is followed by soil strength properties degradation. Note that this behavior was observed for samples loaded with cyclic deviator stress less than 5.3 kPa. The abovementioned mechanism leads to the conclusion that the high amplitude of cyclic loading has a less damaging effect on cohesive material for such loading cycles.

For samples loaded with a high deviator stress level (B.04, C.01, and C.02), we recognize the same response to cyclic loading, i.e., Δ*u_max_* in the first cycle, and then increase of excess pore water pressure at a moderate rate. For samples consolidated in *σ’_c_* equal to 45 kPa, we note the change of Δ*u_max_* characteristics in both cases after 3000 cycles. Furthermore, after this event, the soil again changes the characteristics of Δ*u_max_* (sample C.02), and therefore the pore pressure rises again.

The cohesive material under long-term stress-controlled cyclic loading undergoes a series of hardening and softening stages. Note that these measurements were performed at the top and bottom of the sample. We decided to run additional tests on the model cohesive material which was composed of kaolin minerals. The results for this type of cohesive soil are presented in Figure 5d. For the additional tests, we installed a mid-plane pore pressure transducer. Therefore, the following conclusions can be drawn: The first conclusion from this test is that the pore pressure value in the middle of the sample differs significantly from the pore pressure measured on the top and bottom. The second conclusion is that the characteristic of pore pressure at the middle of the does not correspond with the pore pressure at the ends. For sample D.01, the pore pressure indicates that after 70 cycles, the pore pressure characteristics reverse, but in the middle of the sample, we observe it after 120 cycles and with significantly less scale.

Additionally, pore pressure characteristics again reverse their course in the middle of the specimen, which is not observed at the sample ends. For samples D.03 and D.04, the pore pressure characteristics indicate that the Δ*u* in the middle behaves completely different from the pore pressure registered at the sample ends. The pore pressures at the sample ends show a 10 kPa less excess pore water pressure value caused by cyclic loading and a Δ*u* decrease at a constant rate. Meanwhile, in the middle of the sample, the pore pressure rises in 200 cycles, and then starts to decrease. Both the rate and the value of the pore pressure do not correspond with the pore pressure measured at the sample ends. This non-uniform pore pressure distribution could be caused by the compaction technique, in which the soil is compacted in one direction, and the measurement of pore pressure at the end of samples has a vertical direction, whereas the measurement with midplane pore pressure transducer has a horizontal direction of measurement. Another reason for such differences in measurements is the delay in soil response due to low soil permeability. The differences in the pore pressure value measurements need to be further studied.

The results of this comparison show that the generated pore pressure in various parts of the sample can have different values. The reason for that is the anisotropic nature of the soil in which there are regions more susceptible to the damaging effect of cyclic loading.

### 3.2. Effect of Axial Strain

For cohesive material, the soil response to cyclic loading is one of the most critical factors affecting the construction and which can lead to serviceability failure. If the serviceability limit is crossed, the additional reinforcement must be applied. Figure 6a–d presents the relationship between the axial vertical strain and the number of cycles.

The maximum axial strain in the following cycles *ε^T,N^_a_* characteristics can be divided basically into two categories, i.e., steady and rapid response. The steady response to cyclic loading has a low plastic strain accumulation rate value. The rapid response to cyclic loading is the opposite, the major strain accumulation takes place in the first 1000 cycles. Note that in all cases of steady response, the soil has characteristics for abation reaction to cyclic loading. The rapid response curves are changing their course, and a decrease of the rate can be observed after 1000 cycles. There is no direct relationship between the pore pressure and axial strain. The only distinct connection between Δ*u_max_* and *ε^T,N^_a_* is the high value of excess pore water pressure and low effective consolidation stress *σ’_c_*. The pore water pressure curves for a rapid response seems to reverse the trend after 1000 cycles (for the A.02 sample, after 100 cycles).

To study strain accumulation more extensively, we calculated the value of axial strain rate. The results of the calculations are present in Figure 7a. The strain rate has linear characteristics on the log-log plot. Therefore, strain development has the characteristics of abation for this type of cohesive material. The degradation of Young modulus calculated for the maximum strain in the cycle is presented in Figure 7b.

The plot indicates that the soil stiffness degradation can be calculated based on a well-known model, such as Hardin and Drnevich’s [39] hyperbolic normalized modulus reduction model expressed as Equation (4):(4)$$\frac{E_{N}}{E_{max}}=\frac{1}{1+\left(\varepsilon_{a}^{T,N}/\varepsilon_{a,\;r}^{T}\right)}$$
where the $\varepsilon_{a,\;r}^{T}$ is the axial reference strain ($\varepsilon_{a,\;r}^{T}=q_{max}/E_{max}$).

For such a formula, one can calculate the strain based on the information about initial soils stiffness and the applied cyclic deviator stress *q_max_*, therefore, the abation is connected with stiffness reduction. The stiffness reduction characteristic is useful to characterize the strain development, but the rate of strain remains unclear. Note that, as shown in Figure 7b, the reduction curves have different lengths which do not depend on the initial *E/E_max_* quotient. In addition, as was stated before, the cohesive material response to long cyclic loading is time dependent, and by analysis of Figure 7a, the relationship is quite straightforward. This relationship can be modeled in terms of the number of cycles. However, this method has its limitations. The limitations come from the model constants, which have to be experimentally estimated.

The ideal solution would be the elimination of the time or number of cycles from the analysis. Instead, it would be more fruitful to model the rate of strain accumulation in terms of the ongoing degradation during cyclic loading.

The modulus degradation for searched strain shows how much repetitions one needs to achieve the limit strain or deformation. According to the Janbu’s [40,41] proposed resistance concept, strain accumulation during cyclic loading is unified with the action-reaction analysis, where the action is the constant-stress cyclic loading and the reaction is the strain accumulation. The cohesive material possesses resistance against a forced change of the equilibrium conditions. The resistance concept in the field of cyclic loading [41] can be represented by the following Equation (5):(5)$$R=\frac{dN}{d\varepsilon }$$

The resistance *R* versus number of cycles gives a straight line, where the slope is the creep number *r_s,_* and the resistance can be calculated as Equation (6):(6)$$R=r_{s}N$$

By combining Equations (5) and (6) and integration from *N* = 1 to *N* the cumulative strain *ε^T^_a,cu_* can be calculated [41] based on Equation (7):(7)$$\varepsilon_{a,cu}^{T}=\frac{1}{r_{s}}\ln\left(N\right)$$

This elegant solution has one flaw. The assumption that the resistance characteristics are linear from the beginning of the cyclic loading is a reasonable simplification. However, after the integration, logarithmic characteristics are susceptible to deviations from linear characteristics at the beginning of the tests. During the cyclic loading tests, the logarithmic dependence described by abation is easy to notice. Therefore, we propose a new formula where a logarithm of the number of cycles is related to strain as Equation (8):(8)$$log_{\nu }\left(N\right)=\varepsilon_{a,N}^{T}$$
where the *ν* is the logarithm base which value is the number of cycles to reach $\varepsilon_{a,N}^{T}=1$ and the value of *ν* is the maximum number of cycles *N_max_*. The maximal number of cycles is a large number, and a more useful relationship presents Equation (9):(9)$$\varepsilon_{a,N}^{T}=log_{N_{max}}\left(N\right)=\frac{log_{10}\left(N\right)}{log_{10}\left(N_{max}\right)}$$

The value of the maximal number of cycles changes rapidly at the beginning of the test; therefore, the number of cycle analysis may be subject to a significant error. Instead of calculating the total strain, the cumulative strain can be estimated by Equation (10):(10)$$\varepsilon_{a,N}^{T}=\varepsilon_{a,1}^{T}+\varepsilon_{a,cu}^{T}=\frac{log_{10}\left(N\right)}{r_{\nu }}+\varepsilon_{a,1}^{T}$$
where *r_ν_* is the abation number and *r_ν_* can be calculated based on Equation (11):(11)$$r_{\nu }=\frac{log_{10}\left(N\right)}{\varepsilon_{a,N}^{T}-\varepsilon_{a,1}^{T}}$$

For practical applications, the abation number value can be calculated for cycle *N* = 10,000. After this cycle, the *r_ν_* value has an almost constant value. To calculate the abation number, we conducted multilinear regression analysis which led us to establish the relationship between the soil stress and stiffness state using the following formula Equation (12):(12)$$r_{v}=696.548+0.595{\sigma^{\prime }}_{3}+6.207q_{a}+0.916q_{m}+321.14CSR-487.98e_{0}-382.6\varepsilon_{a,\;1}^{T}\;+22.69E_{1}-1.184E_{max}-5232,45\frac{E_{1}}{E_{max}}-1,214,512.5\varepsilon_{a,\;max}^{T}$$

For the presented formula, the coefficient of determination R^2^ is equal to 0.908. In Appendix A, we present the details concerning strain resistance analysis.

### 3.3. The Cumulative Pore Pressure Resistance

The pore pressure characteristics, presented in Figure 3, show that the excess pore pressure generation has different courses during cyclic loading, and its value is different across the sample. The concept presented by Janbu for resistivity also considers the cumulative pore pressure resistance *R_u_* as Equation (13):(13)$$R_{u}=∆q\frac{dN}{du}$$
where Δ*q* is the double deviator stress amplitude. According to Equation (10) and the soil strain resistance characteristics, with the following equation, the excess pore pressure can be calculated as Equation (14):(14)$$∆u_{cu}=\frac{∆q}{r_{u}}\ln\left(N\right)$$
where *r_u_* is the pore pressure resistance. During the data analysis, we decided to omit the Δ*q* and to simplify the relationship presented above. The new relationship between the excess pore water pressure and the number of cycles is as Equation (15):(15)$$∆u_{cu}=\frac{1}{r_{u}}\ln\left(N\right)+∆u_{cu,1}$$
where the pore pressure resistance is calculated as *r_u_* = *dN/du* and Δ*u_cu_* is the excess pore pressure in the first cycle of loading. Based on the test result analysis, the Δ*u_cu,_* value can be calculated using the relationship between the initial isotropic consolidation pressure *p’* and the CSR value as Equation (16):(16)$${p^{\prime }}_{1}={p^{\prime }}_{0}-{p^{\prime }}_{0}\cdot CSR$$
where *p*′_1_ = *p*′_0_ − Δ*u_cu,_*_1_ is the effective mean stress in the first cycle. This relationship is presented in Figure 8, where we present the relationship between the calculated and measured *p*′ value.

The cyclic triaxial test results show that the soil generates excess pore water pressure until a certain point where we observe the reverse of pore pressure versus the number of cycles characteristic. Therefore, some limit to the pore pressure generation needs to be introduced. The Arulanandan et al. [34] creep mechanism proposal can also be adopted in the case of the cyclic loading subject. The limit value can also be expressed as the limit mean effective stress *p*′*_mim_* value. For the calculation of *p*′*_mim_*, we use the Modified Cam-Clay yield surface Equation (17):(17)$$0=q_{max}{}^{2}+M^{2}\;p^{\prime }\left({p^{\prime }}_{c}-p^{\prime }\right)$$
where *p*′*_c_* is the consolidation pressure and *M* is the slope of the critical state line. During one-way cyclic loading tests, the deviator stress maximal value is constant (*q_max_* = *const.*), as well as the slope of critical state line CSL. Therefore, *p*′*_min_* is the point of excess pore pressure reverse, where the sample starts to behave unstably, and the yield surface shrinks to the point on the critical state line. The *dN/du* value versus number of cycles characteristics show a specific mechanism that occurs when the *p*′*_miní_* is achieved. This mechanism indicates that beyond this point, the sign of the *r_u_* changes, but the value remains constant. The schema of this relationship is presented in Figure 9a. The *dN/du* absolute value against the number of cycles *N* is presented in Figure 9b.

This behavior is favorable for excess pore pressure modeling. After the *p*′*_min_* point of effective stress, the pore pressure decreases, and the unstable state lasts until the soil reaches failure point, which is the critical state point defined as Equation (18):(18)$${p^{\prime }}_{f}=\frac{q_{max}}{M}$$

Therefore, based on the Equations (17) and (18), one can calculate the total number of cycles to failure *N_f_* and the total strain at failure $\varepsilon_{a,N_{f}}^{T}$ using Equation (10).

### 3.4. Failure Analysis

In this section, we show the calculations for the sample D.01. The calculations were conducted for the pore pressure resistance *r_u_* equal to 0.5579, and the abation number *r_v_* equal to 91.4. The slope of the critical state line *M* is equal to 0.873 (for φ′ = 22.4°), and the preconsolidation pressure *p*′*_c_* is set to 270 kPa. Therefore, based on Equation (17), the minimal effective pore pressure *p*′*_min_* is equal to 12.85 kPa, which indicates that the maximal pore pressure Δ*u_max_* reaches 32.15 kPa, after the Δ*u_max_* poilt, the excess pore water pressure decreases. The failure point in these conditions is defined by Equation (18), where for these conditions, the *p*′*_f_* equals to 49.02 kPa. Since the pore pressure development during cyclic loading is governed by a constant pore pressure resistance, one can calculate the number of cycles required to obtain the *p*′*_f_* value. For sample D.01, the required number of cycles, *N_f_*, is equal to 1.13 × 10^10^. Finally, we can calculate the strain at failure based on Equation (10) which gives $\varepsilon_{a,N_{f}}^{T}$ equal to 0.1272 (12.72%). Figure 10 presents the results of the calculations versus the number of cycles.

Analysis of Figure 8 shows that a significant number of cycles are required to achieve the critical state in these conditions. Therefore, the unstable state after the soil reaches *p*′*_min_* means gradual soil softening, which results in preconsolidation pressure degradation during long-term cyclic loading.

## 4. Damage Mechanism Considerations

The presented soil failure analysis is able to calculate the number of cycles to failure using a modified resistance concept. Nevertheless, failure analysis using *N_f_* needs to account for three kinds of failure, which are called plastic collapse; ratcheting, which is defined as a loss in functionality due to excessive deformation; and collapse due to low cycle fatigue. Low cycle fatigue can only occur in symmetric stress cycling, which is not considered in this article. Plastic collapse is simply the load which leads to failure due to excessive pore water pressure generation during cyclic loading. The sample reaches yield surface in the first cycle, and since the load is high enough that further cycles will not contribute to excess pore water pressure generation, but to an increase in the effective pore water pressure. The cyclic loading lasts a few cycles, during which a soil sample reaches a critical state point. The yield surface shrinkage during cyclic loading was also noticed by Ni et al. [35].

Ratcheting is the state when the load is significant enough to cause an increase in excess pore pressure and, after a few hundred cycles, an unstable soil state. This type of behavior is similar to that which was presented as a cyclic creep mechanism. Ratcheting behavior can occur as three types. Plastic shakedown is when a sample behavior shows a decrease in strain rate to the point where no additional strain is noticed. Constant ratcheting occurs when one observes a constant plastic strain rate during cyclic loading. The last type of ratcheting behavior is increasing ratcheting strain rate, which eventually leads to plastic collapse. This description is valid for continuous materials such as steel or concrete. The concept of resistance shows that the soil plastic strain rate always decreases. Therefore, for example, when the consolidation stress is high as well as the soil resistance, plastic shakedown occurs simply because the decrease of strain rate is rapid, the number of cycles to failure is much higher than the service life of the construction which indicates infinite life. The failure mechanism is held not by strain development but by excess pore pressure development. In all analyzed cases, the strain abation resulted in plastic shakedown if the effective stress path did not lead to the first unstable state, and then to failure by reaching the critical state.

When excess pore water pressure causes an unstable state, a sample has also undergone plastic shakedown, but during this process, preconsolidation pressure decreases and, supposedly, the strength parameters as well. The above-mentioned mechanism is presented in Figure 11.

The recently developed soil constitutive models that capture a cohesive soil response to cyclic loading are often based on more than one yield surface solution [35,42,43]. This approach is correct when a short-term cyclic loading is considered. For long-term cyclic loading, the phenomena of creep, abation, and fatigue need to be accounted for.

## 5. Conclusions

Four series of cyclic triaxial tests in undrained conditions on four types of cohesive soil provide general knowledge about soil behavior under cyclic loading. Such knowledge is vital for the study of the damaging effect. The development of foundation design procedures or gravity structures requires detailed knowledge about possible failure scenarios for the ultimate and serviceability limit analysis. On the basis of the test result analysis, we can draw the following conclusions:Cyclic loading causes a build-up of excess pore water pressure. The Δ*u* characteristic in the number of cycles argument, in the case of a few samples, leads to the observation of the peak pore pressure Δ*u_max_*, and then reverses of excess pore pressure characteristics. This phenomenon leads to an undrained creep where Δ*u* decreases, and the effective stress path touches the yield surface.The cohesive material under long-term stress-controlled cyclic loading undergoes a series of hardening and softening stages. The results of cyclic triaxial tests with an installed mid-plane pore pressure transducer indicate that the pore pressure value in the middle of the sample differs significantly from the pore pressure measured on the top and bottom of the sample. In addition, the pore pressure at the middle of the sample characteristic does not correspond with the pore pressure at the ends. Therefore, further studies are necessary to understand pore pressure behavior during cyclic loading.Analysis of the strain rate shows that this relationship has a linear characteristic on the log-log plot, and therefore strain development has abation characteristics for this type of cohesive material. We observed soil stiffness degradation, which were calculated based on a well-known model such as Hardin and Drnevich’s [39] hyperbolic normalized modulus reduction model. To model the rate of strain, we used Janbu’s proposal for the resistance concept. Strain accumulation during the cyclic loading is governed by the action-reaction analysis, where the action is the constant-stress cyclic loading and the reaction is the strain accumulation, and it is an easy tool to track the number of cycles during stiffness degradation.The abation characteristics during the cyclic loading tests have logarithmic dependence from the number of cycles. Therefore, we propose a new formula where a logarithm of the number of cycles is related to strain, and we estimated the *ν* parameter which is the logarithm base of the number of cycles to reach $\varepsilon_{a,N}^{T}=1$ and the value of *ν* is the maximum number of cycles *N_max_*.To model the excess pore water pressure generation, we propose a limit, which incorporates the Arulanandan et al. creep mechanism proposal, on the cyclic loading subject. The limit value is limit mean effective stress *p*′*_mim_*, and for its calculation, we use the Modified Cam-Clay yield surface equation. The *p*′*_min_* is the point of excess pore pressure reverse, where the sample starts to behave unstably, and the yield surface shrinks to a point on the critical state line.When the stress path achieves the *p*′*_min_* value, the *dN/du* versus number of cycles characteristics show specific phenomena where the sign of the *r_u_* changes, but the value remains constant, and excess pore water pressure decreases.For the presented mechanism, we discuss the possible effect of cyclic loading damaging effects in undrained conditions.It is worth mentioning that fatigue testing of the materials is loading-path dependent. In this study, first, the cohesive material was compressed, and then we applied cyclic shear loading. The loading path impacts fatigue assessment, as well as the geometrical configuration and material properties [44,45]. To provide a more holistic understanding of fatigue failure, more loading types needed to be considered, for instance, cyclic circumferential stress and cyclic shear stress in a hollow cylinder apparatus.

## Figures and Tables

**Figure 1 materials-13-03907-f001:**
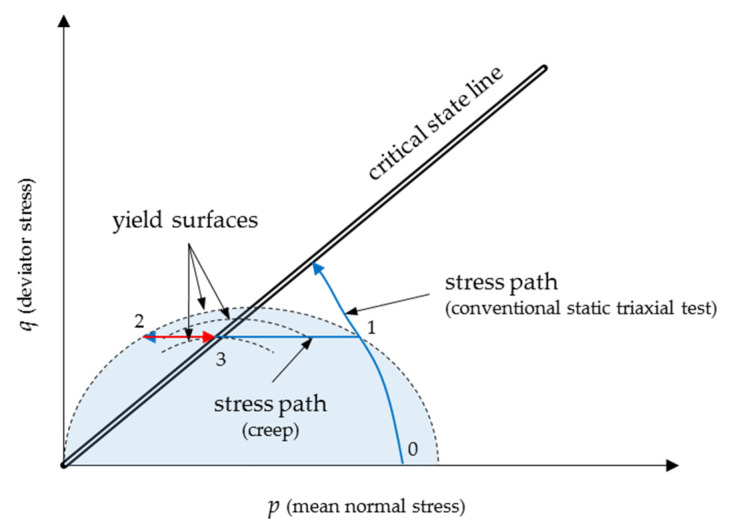
The undrained static creep failure in the scope of the critical state concept.

**Figure 2 materials-13-03907-f002:**
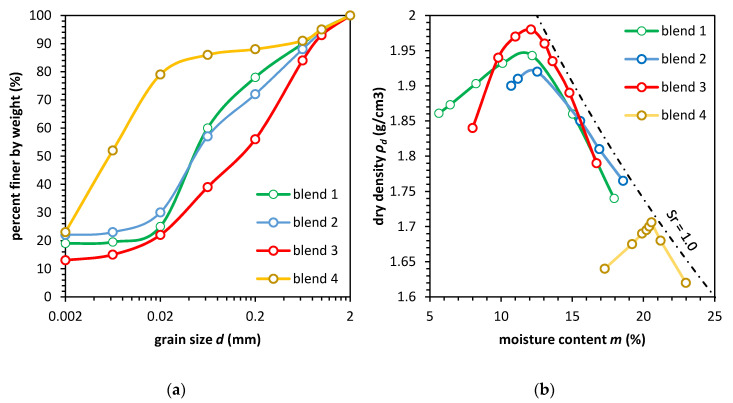
The physical properties of cohesive material in this study. (**a**) Soil gradation curve; (**b**) Optimal moisture test results following the Proctor method for four soil types.

**Figure 3 materials-13-03907-f003:**
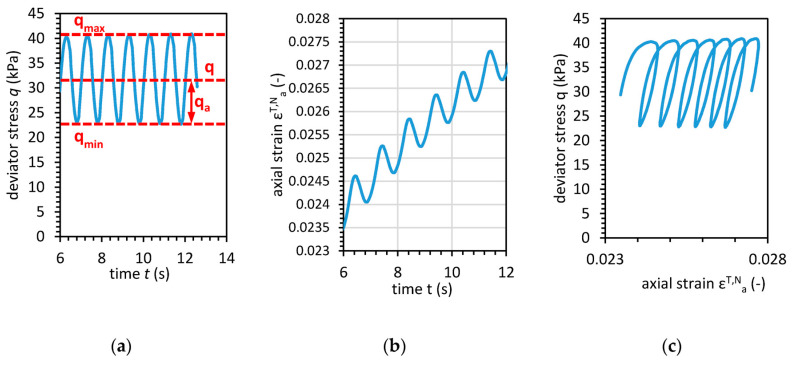
The load conditions of cohesive material during the cyclic triaxial test. (**a**) Deviator stress-time relationship; (**b**) Axial strain-time relationship; (**c**) Deviator stress-axial strain characteristics.

**Figure 4 materials-13-03907-f004:**
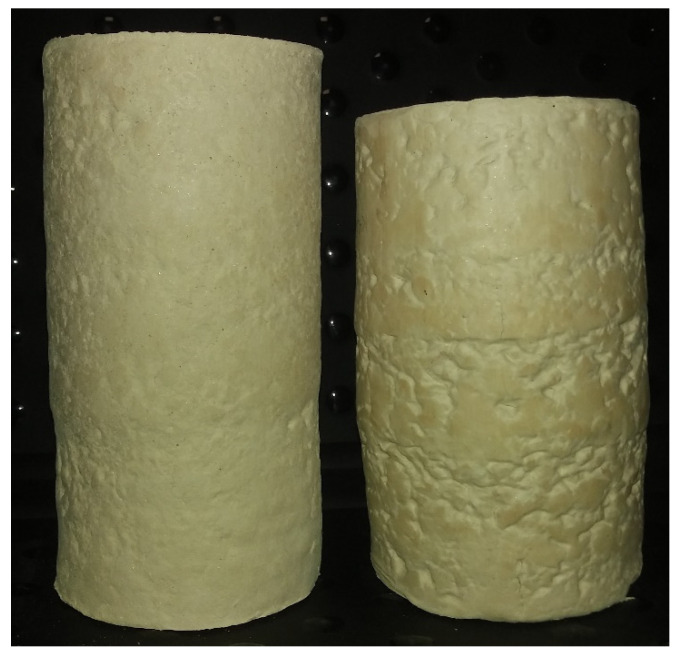
Image of pre- and post-testing of the samples of cohesive material.

**Figure 5 materials-13-03907-f005:**
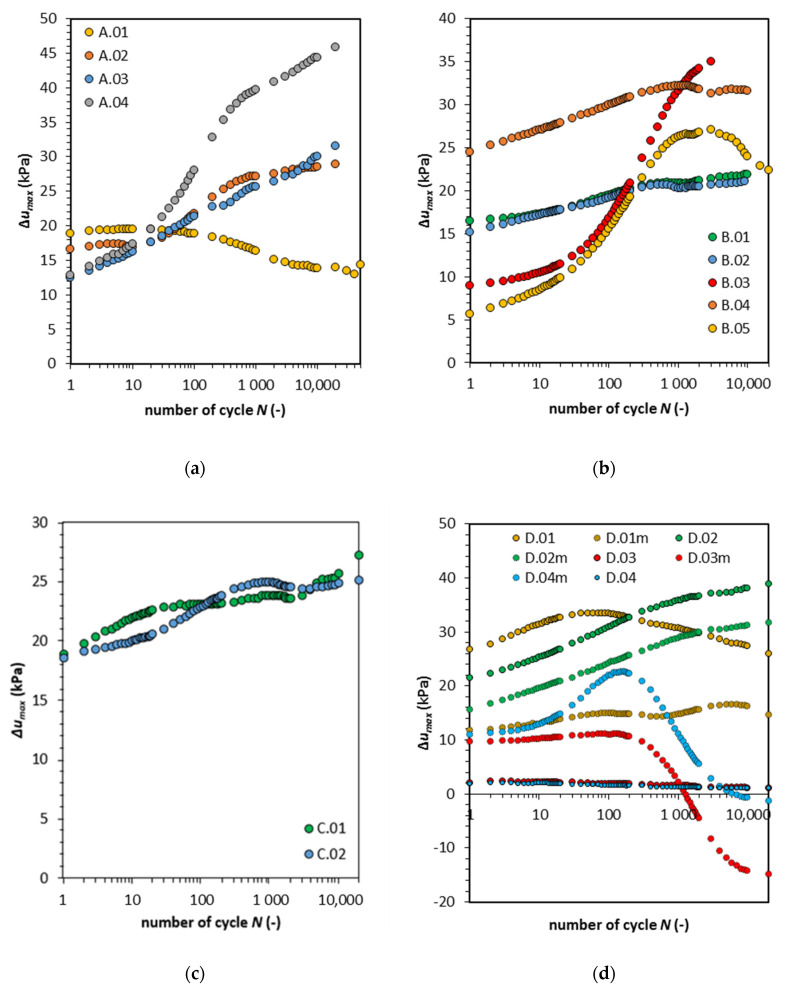
Maximum excess pore water pressure versus the number of cycles. (**a**) Type 1; (**b**) Type 2; (**c**) Type 3; (**d**) Type 4.

**Figure 6 materials-13-03907-f006:**
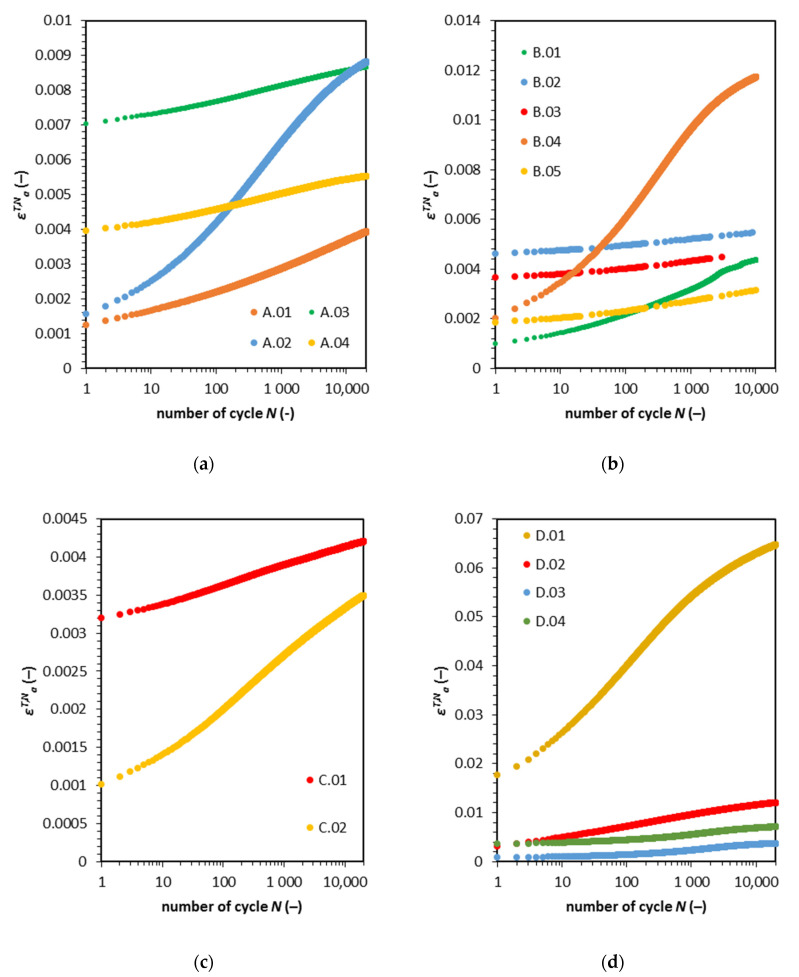
Maximum axial strain versus the number of cycles. (**a**) Type 1; (**b**) Type 2; (**c**) Type 3; (**d**) Type 4.

**Figure 7 materials-13-03907-f007:**
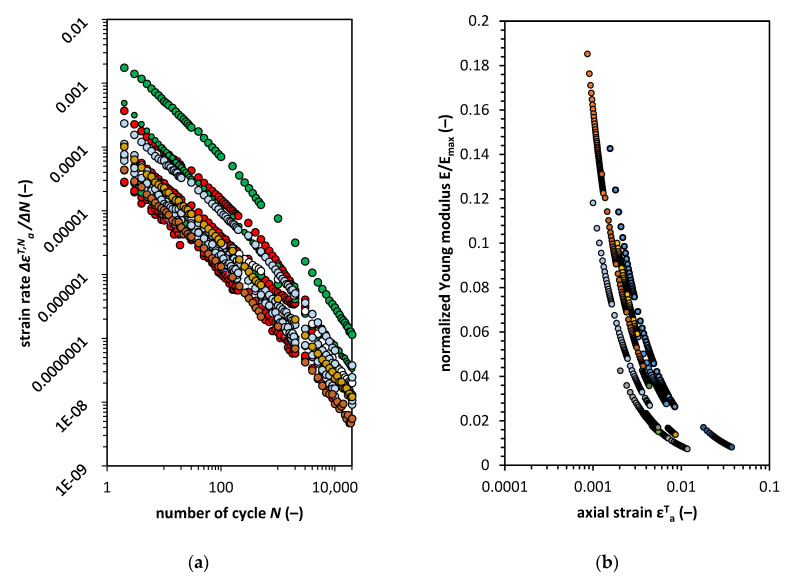
(**a**) Strain rate for tested soils for the following number of cycles; (**b**) normalized Young modulus characteristic versus maximal axial strain for samples loaded with cyclic force.

**Figure 8 materials-13-03907-f008:**
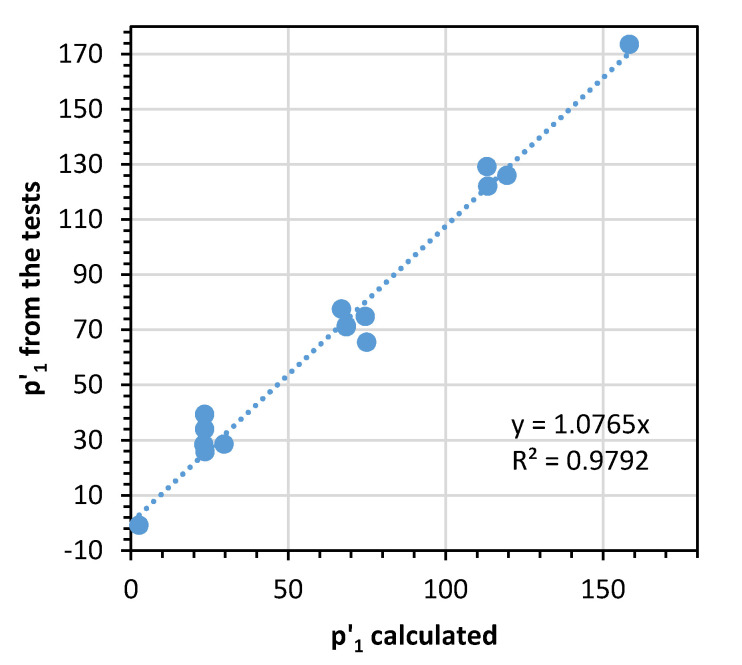
The relationship between the mean effective stress in the first cycle calculated based on Equation (16) and calculated form the cyclic triaxial tests.

**Figure 9 materials-13-03907-f009:**
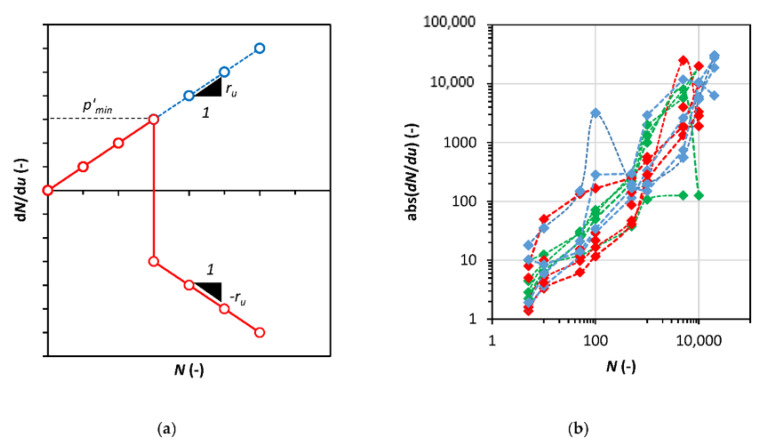
(**a**) Mechanism of pore pressure development during cyclic loading; (**b**) Test results for the *dN/du* characteristics from cyclic triaxial tests.

**Figure 10 materials-13-03907-f010:**
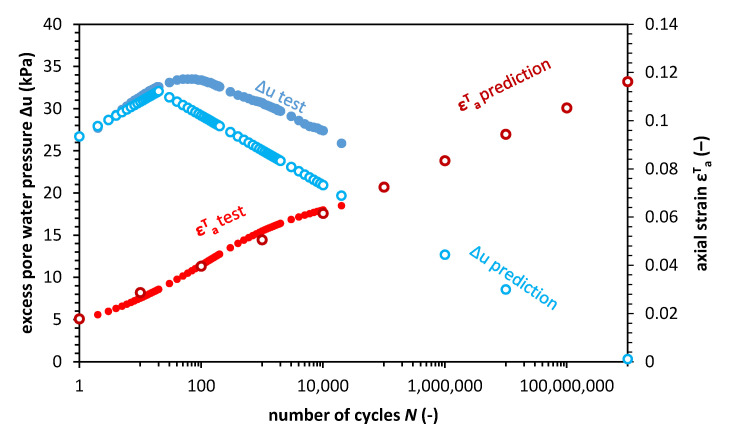
Excess pore pressure and axial strain test results and results of calculation with the prognosis for sample D.01.

**Figure 11 materials-13-03907-f011:**
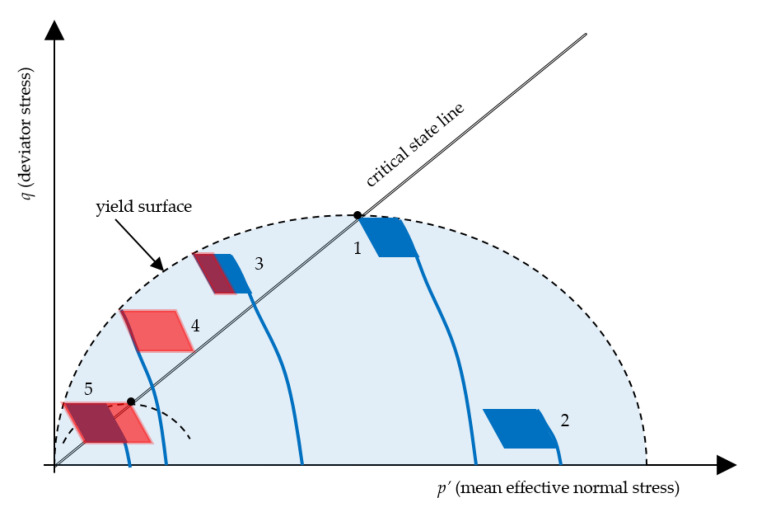
Schema of cyclic loading stress paths in undrained conditions (*e* = const.). The following numbers indicate: (**1**) Plastic failure due to excessive loading; (**2**) Plastic shakedown; (**3** and **4**) Plastic shakedown in an unstable state; (**5**) Ratcheting. Black dots indicate failure.

**Table 1 materials-13-03907-t001:** Physical properties of four types of cohesive material tests in this study.

	Type 1	Type 2	Type 3	Type 4
Soil Classification	sasiCl	sasiCl	clSa	siCl
W_P_ (%)	23.6	15.2	13.5	24.4
W_L_ (%)	42.0	38.7	23.4	42.4
I_P_ (%)	8.4	14.5	6.9	18.0
A_C_ (–)	0.44	0.67	0.53	0.78
*φ’* (°)	28.1	26.7	30.0	22.4
*c* (kPa)	6.8	9.8	5.2	20.5

W_P_—plasticity limit, W_L_—liquid limit, I_P_—plasticity index, A_C_—colloidal activity, *φ’*—effective friction angle, *c—*cohesion.

**Table 2 materials-13-03907-t002:** Cyclic triaxial loading program of cohesive materials.

	Sample Number	*Σ’_C_* (kPa)	*Q_A_* (kPa)	*Q_M_* (kPa)	*CSR* (–)	*N* (–)	*E_0_* (–)	*P_D_* (G/CM^3^)	*M* (%)
**Type 1**	A.01	18.0	3.9	26.9	0.856	5 × 10^4^	0.407	1.897	14.33
	A.02	45.0	4.1	39.5	0.484	2 × 10^4^	0.402	1.904	14.75
	A.03	90.0	4.5	41.5	0.256	2 × 10^4^	0.345	1.985	12.62
	A.04	135.0	3.95	39.2	0.160	5 × 10^4^	0.371	1.948	13.65
**Type 2**	B.01	45.0	5.2	25.6	0.342	10^4^	0.379	1.936	11.18
	B.02	90.0	5.3	25.7	0.172	10^4^	0.360	1.963	10.71
	B.03	135.0	5.3	25.7	0.115	10^4^	0.357	1.967	15.56
	B.04	45.0	11.0	32.0	0.478	5 × 10^4^	0.412	1.882	13.61
	B.05	90.0	2.75	27.3	0.167	5 × 10^4^	0.380	1.892	14.03
**Type 3**	C.01	90.0	10.8	32.3	0.239	5 × 10^4^	0.361	1.961	12.07
	C.02	45.0	11.0	32.0	0.478	5 × 10^4^	0.375	1.942	12.68
**Type 4**	D.01	45.0	10.2	32.6	0.476	2 × 10^4^	0.639	1.629	19.73
	D.02	90.0	10.8	32.3	0.239	2 × 10^4^	0.601	1.668	22.98
	D.03	135.0	11.0	32.8	0.162	2 × 10^4^	0.598	1.671	20.14
	D.04	180.0	10.6	32.6	0.120	2 × 10^4^	0.607	1.661	17.28

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
