# Peer review of "Long-Term Cyclic Loading Impact on the Creep Deformation Mechanism in Cohesive Materials"

_materials, 2020, doi:10.3390/ma13173907_

Round 1
Reviewer 1 Report
Please refer to the attached file for the reviewer's comments

Author Response
Respond for Review Report 1 for submission of a paper to Materials – MDPI journal
24.07.2020
Dear Reviewer,
We would like to thank You for all Your kind remarks. Please, find below the responses to Comments and Suggestions.
“The main work done in the manuscript is on creep plastic deformation analysis. Please note that in stress/strain analysis with plastic deformation, the simulation results are loading-path dependent: you add tension first then shear loading, or shear loading first then tension, the results will be substantially different. The authors may refer to such work as “Fatigue analysis on offshore pipelines subjected to different loadings”, Ocean Engineering, 117 (2016). 45-56; “Fatigue crack growth investigation on offshore pipelines with three-dimensional interacting cracks”. Geoscience Frontiers, Vol. 9, No.6, 1689 -1698, 2018. Have the authors considered the path-dependent effect in their experimental testing and results analysis?.”
Thank You for this note. Indeed in this study, we applied a specific loading type, and in conclusions, we highlighted that the loading path model has a great impact on fatigue assessment. We also referred to the mentioned literature to strengthen this point. In this article, we did not consider the path-dependent effect, but we are plan to perform the cyclic hollow cylinder tests for fatigue assessment and two-way cyclic triaxial tests.
“The authors should compare their results/findings with those available in open literature. If there are no direct related results available, the authors should study some cases which have been done by other researchers, and compare the results among them. This is for the validation purpose of the experimental testing conducted in the current research.”
We compared the test results with constitutive models for cohesive materials developed in the last decade.
All changes in the manuscript ate highlighted by the red font. We checked the manuscript for typos, and we improved the English language.
Thank You for Your remarks on this manuscript.
Sincerely,
Andrzej and Wojciech
Reviewer 2 Report
The manuscript is not well written, which renders it very difficult to read and understand. In addition, a thorough analysis of the experimental results is lacking. Consequently, the reader is left with a lot of questions. In particular:
- The tests are not well explained in terms of samples, loading conditions and measured parameters.
- Loading conditions:
- stress, pressure, and deviator stress are being used in the text. These are in fact dependent parameters: the stress tensor can be additively decomposed into a deviatoric stress tensor and the hydrostatic stress tensor (or pressure). It is recommended to properly define these parameters in the first part of the manuscript.
- Related to this, please specify cyclic loading conditions, including time scales
- It is suggested to include a schematic of the test which explains these details. Furthermore, pictures of actually used samples, ideally pre- and post-testing, should be included as well.
- Loading conditions:
- Delta U is not introduced in the text, it’s mentioned in Fig.3 as ‘excess pore water pressure’
- In the results discussion sections, please use more quantitative descriptions instead of ‘high’, ‘low’, etc.
- Contrary to the text, Table 2 does not include all information about the cyclic loading program details
- Section 3.1: it is remarked that the pore pressure is not uniform in the samples. This is interesting and should be studied in more detail. Microscopic analysis of cross-sectioned samples could be a means to achieve this. In addition, it would be interesting to analyze the origin of these non-uniformities instead of only reporting these. What is the effect of sample manufacturing (e.g., imperfect initial sample geometry and non-uniform microstructure)? Finally, it would be useful to illustrate the distribution of the pore pressure over the height/length of the sample.
- Long-term degradation mechanisms have been studied for polymers and metals as well (fatigue and creep mechanisms). Detailed analyses have been reported in literature for several decades. Do the authors think that these rather mature models and experimental results could be relevant for their cohesive soil materials?
- Section 1 – why is Perzyna’s model mentioned without actually applying it?
- Finally, the manuscript contains various grammar and spelling mistakes. These should be corrected.
Author Response
Respond for Review Report 2 for submission of a paper to Materials – MDPI journal
24.07.2020
Dear Reviewer,
We want to thank the Reviewer 2 for all remarks. Below are the responses to Comments and Suggestions.
“Loading conditions: stress, pressure, and deviator stress are being used in the text. These are in fact dependent parameters: the stress tensor can be additively decomposed into a deviatoric stress tensor and the hydrostatic stress tensor (or pressure). It is recommended to properly define these parameters in the first part of the manuscript.”
We added the subchapter where we present the stress parameters derived from stress tensor, as well as the loading conditions.
“Related to this, please specify cyclic loading conditions, including time scales It is suggested to include a schematic of the test which explains these details. Furthermore, pictures of actually used samples, ideally pre- and post-testing, should be included as well.”
The required by Reviewer parts including load conditions and pictures of samples was added to the manuscript.
“Delta U is not introduced in the text, it’s mentioned in Fig.3 as ‘excess pore water pressure’”
We introduced the excess pore water pressure definition.
“In the results discussion sections, please use more quantitative descriptions instead of ‘high’, ‘low’, etc.”
We changed the descriptions to more quantitative as reviewer suggested.
“Contrary to the text, Table 2 does not include all information about the cyclic loading program details”
The reviewer is right, we replaced the word “detailed”.
“Section 3.1: it is remarked that the pore pressure is not uniform in the samples. This is interesting and should be studied in more detail. Microscopic analysis of cross-sectioned samples could be a means to achieve this. In addition, it would be interesting to analyze the origin of these non-uniformities instead of only reporting these. What is the effect of sample manufacturing (e.g., imperfect initial sample geometry and non-uniform microstructure)? Finally, it would be useful to illustrate the distribution of the pore pressure over the height/length of the sample.”
This non-uniform pore pressure distribution may be a cause of the compaction technique where the soil is compacted in one direction and the measurement of pore pressure at the end of samples have a vertical direction while the measurement with mid plane pore pressure transducer have a horizontal direction of measurement. Another reason for such difference in measurements is the delay in soil response due to a low soil permeability. We want to thank the reviewer for this suggestion and we are going to explore this matter in further papers.
“Long-term degradation mechanisms have been studied for polymers and metals as well (fatigue and creep mechanisms). Detailed analyses have been reported in literature for several decades. Do the authors think that these rather mature models and experimental results could be relevant for their cohesive soil materials?”
The long-term cyclic degradation mechanism in field of polymers and metals is indeed advanced and the literature covering this matter is very reach. In field of soil mechanics the cyclic loading problem is rather new and bases on the mature mechanisms from the metal studies as for example shakedown concept. We think, that this mature models would be really helpful to understand the fatigue of soil especially from the micro-structure point of view which is poorly developed in soil mechanics.
“Section 1 – why is Perzyna’s model mentioned without actually applying it?”
We decided to study the strain and pore pressure development with the resistance concept which reflects well the abation and creep respond to cyclic loading despite the advantages of Przyna’s visco-plascity model.
“Finally, the manuscript contains various grammar and spelling mistakes. These should be corrected.”
We checked the manuscript for typos, and we improved the English language.
All changes in the manuscript ate highlighted by the red font.
Thank You for Your remarks on this manuscript.
Sincerely,
Andrzej and Wojciech
Reviewer 3 Report
This paper is on a topic of wide interest in geotechnical engineering. The authors have a done a fairly good review of the current literature on the models used. Their use of Janbu’s model is appropriate.
However, in their discussion of critical state model for cyclic loading they have failed to note recent advances in modelling to the concepts highlighted on Figures 1 and Figure 9. They should review and cite the work by the following authors on cyclic loading interpretation.
Jing Ni; Buddhima Indraratna, ; Xue-Yu Geng; John Phillip Carter and You-Liang Chen. Model of Soft Soils under Cyclic Loading, Int. J. Geomech., 2015, 15(4): 04014067.
The paper needs improvement in English.
Author Response
Respond for Review Report 3 for submission of a paper to Materials – MDPI journal
20.07.2020
Dear Reviewer,
We want to thank the Reviewer 3 for all remarks. Below are the responses to Comments and Suggestions.
“However, in their discussion of critical state model for cyclic loading they have failed to note recent advances in modelling to the concepts highlighted on Figures 1 and Figure 9. They should review and cite the work by the following authors on cyclic loading interpretation.
Jing Ni; Buddhima Indraratna, ; Xue-Yu Geng; John Phillip Carter and You-Liang Chen. Model of Soft Soils under Cyclic Loading, Int. J. Geomech., 2015, 15(4): 04014067.
The paper needs improvement in English.”
The mentioned paper about Cam-Clay constitutive model in cyclic loading is indeed a significant improvement to the literature review. We discussed the model in the manuscript.
All changes in the manuscript ate highlighted by the red font. We checked the manuscript for typos, and we improved the English language.
Thank You for Your remarks on this manuscript.
Sincerely,
Andrzej and Wojciech